# Altitude Effect on Cutaneous Melanoma Epidemiology in the Veneto Region (Northern Italy): A Pilot Study

**DOI:** 10.3390/life12050745

**Published:** 2022-05-17

**Authors:** Paolo Del Fiore, Irene Russo, Alessandro Dal Monico, Jacopo Tartaglia, Beatrice Ferrazzi, Marcodomenico Mazza, Francesco Cavallin, Saveria Tropea, Alessandra Buja, Rocco Cappellesso, Lorenzo Nicolè, Vanna Chiarion-Sileni, Chiara Menin, Antonella Vecchiato, Angelo Paolo Dei Tos, Mauro Alaibac, Simone Mocellin

**Affiliations:** 1Soft-Tissue, Peritoneum and Melanoma Surgical Oncology Unit, Veneto Institute of Oncology IOV-IRCCS, 35128 Padua, Italy; irene.russo@phd.unipd.it (I.R.); marcodomenico.mazza@iov.veneto.it (M.M.); saveria.tropea@iov.veneto.it (S.T.); antonella.vecchiato@iov.veneto.it (A.V.); simone.mocellin@unipd.it (S.M.); 2Division of Dermatology, Department of Medicine (DIMED), University of Padua, 35128 Padua, Italy; alessandrodalmonico@gmail.com (A.D.M.); jacopo.tartaglia@studenti.unipd.it (J.T.); mauro.alaibac@unipd.it (M.A.); 3Postgraduate School of Occupational Medicine, University of Verona, 37129 Verona, Italy; ferrazzi.beatrice@gmail.com; 4Independent Statistician, 36020 Solagna, Italy; cescocava@libero.it; 5Department of Cardiological, Thoracic, Vascular Sciences and Public Health, University of Padua, 35128 Padua, Italy; alessandra.buja@unipd.it; 6Pathological Anatomy Unit, University Hospital of Padua, 35128 Padua, Italy; rocco.cappellesso@gmail.com (R.C.); angelo.deitos@unipd.it (A.P.D.T.); 7Unit of Pathology & Cytopathology, Department of Medicine (DIMED), University of Padua, 35128 Padua, Italy; lorenzo.nick86@gmail.com; 8Unit of Surgical Pathology & Cytopathology, Ospedale dell’Angelo, 30174 Mestre, Italy; 9Melanoma Oncology Unit, Veneto Institute of Oncology IOV-IRCCS, 35128 Padua, Italy; vanna.chiarion@iov.veneto.it; 10Immunology and Diagnostic Molecular Oncology Unit, Veneto Institute of Oncology IOV-IRCCS, 35128 Padua, Italy; chiara.menin@iov.veneto.it; 11Department of Surgery, Oncology and Gastroenterology (DISCOG), University of Padua, 35128 Padua, Italy

**Keywords:** cutaneous melanoma, altitude, coast-plain-hill gradient

## Abstract

The incidence of cutaneous melanoma has been increasing in the last decades among the fair-skinned population. Despite its complex and multifactorial etiology, the exposure to ultraviolet radiation (UVR) is the most consistent modifiable risk factor for melanoma. Several factors influence the amount of UVR reaching the Earth’s surface. Our study aimed to explore the relationship between melanoma and altitude in an area with mixed geographic morphology, such as the Veneto region (Italy). We included 2752 melanoma patients who were referred to our centers between 1998 and 2014. Demographics, histological and clinical data, and survival information were extracted from a prospectively maintained local database. Head/neck and acral melanoma were more common in patients from the hills and the mountains, while limb and trunk melanoma were more common in patients living in plain and coastal areas. Breslow thickness, ulceration and mitotic rate impaired with increased altitude. However, the geographical area of origin was not associated with overall or disease-free survival. The geographical area of origin of melanoma patients and the “coast-plain-hill gradient” could help to estimate the influence of different sun exposure and to explain the importance of vitamin D levels in skin-cancer control.

## 1. Introduction

In the last decades, the incidence of melanoma has been continuously increasing around the world [1,2]. In Italy, melanoma incidence and mortality display a large variation across the country, and the geographic variability is associated with a decreasing incidence from Northern (22 cases/100,000 people) to Southern Italy (about 10 cases/100,000 people) [3,4]. In the Veneto region (North-eastern Italy), melanoma cases have more than tripled in the last 30 years, with a heterogeneous incidence within the regional area [5].

The epidemiology of melanoma is complex, and individual risk depends on the patient, genetic and environmental risk factors, as well as their interactions [6]. The most important and potentially modifiable environmental risk factor for developing melanoma is the excessive exposure to ultraviolet radiation (UVR) due to the genotoxic effect. Several factors influence the amount of UVR reaching Earth’s surface, including latitude, altitude, ozone depletion, UV light elevation and weather conditions [7].

Among factors associated with UVR exposure, altitude has been estimated to contribute to a 10−12% emission increment for every 1000 m of elevation [7,8,9,10,11]. 

The Veneto region is characterized by a mixed morphology including hills (15%), mountains (29%), and plains or coastal areas (56%), with altitudes ranging from sea level up to 3383 m above sea level, but with negligible differences in terms of latitude [12]. Therefore, this seems a suitable area for exploring the relationship between melanoma and altitude within a population of similar pigmentation characteristics.

This study aimed to investigate the clinicopathological characteristics and outcomes in a cohort of melanoma patients living in the Veneto region (Italy) according to the different geographical areas of residency.

## 2. Materials and Methods

### 2.1. Study Design

This is a retrospective cohort study on melanoma patients who were diagnosed and/or treated at the Veneto Institute of Oncology (IOV) and at the University Hospital of Padua (UHP) over a period of 16 years. The study was conducted in accordance with the Declaration of Helsinki principles and all patients gave their consent for data collection and analysis for scientific purposes. The study was approved by the local Ethical Committee (CESC IOV Not.2 on 20 January 2020).

### 2.2. Patients

All patients who were diagnosed and/or treated for melanoma from 1998–2014 at IOV and at UHP (Italy) were considered for inclusion in the study. The inclusion period was chosen to potentially achieve a minimum follow-up of five years at the time of data analysis. Most patients are usually referred to the Veneto Institute of Oncology for diagnosis and/or first-line treatment, while some patients are referred for disease progression after being treated in peripheral centers.

### 2.3. Diagnosis 

The diagnosis of melanoma was histologically confirmed according to the fourth edition of the World Health Organization (WHO) classification of skin tumors [13]. The staging was updated in the 8th edition of the Union for International Cancer Control (UICC) TNM Classification of Malignant Tumours [14].

### 2.4. Data Collection

All data were extracted from a prospectively maintained local database. Data collection included demographics, tumor characteristics, and follow-up information. The residential area was classified according to four geographical categories (hill, mountain, plain, and coast) based on patient address and according to the Italian Central Statistics Institute (ISTAT) [15]. In Northern Italy, the Institute classifies residential areas above 600 m as “mountain”, those within 300–600 m as “hill”, those below 300 m as “plain” and those with sea access as “coast”.

Follow-up information was extracted from scheduled visits. Follow-up duration was calculated from the date of diagnosis to 31 December 2019. Disease-specific survival was calculated from the date of diagnosis to the date of disease-related death (uncensored case) or last visit (or disease-unrelated death) (censored case). Disease-free survival was calculated in patients with primary melanoma from the date of diagnosis to the date of disease recurrence or the date of last visit (or death). Recurrence included local recurrence, regional skin/in-transit metastases, regional lymph node metastases, and/or distant metastases.

### 2.5. Statistical Analysis

Continuous data were summarized as median and interquartile range (IQR). Categorical data were compared between groups using the Chi Square test or Fisher’s exact test, while a Mann–Whitney test and a Kruskal-Wallis test were used for continuous data. Survival curves were calculated using the Kaplan–Meier method and compared using the log-rank test. Multivariable analyses of survival (disease-specific survival and recurrence-free survival) were performed using Cox regression models, and effect sizes were expressed as hazard ratio (HR), with a 95% confidence interval (CI).

Since the participating centers are the hubs for most patients living in the plain area, a sensitivity analysis including only referred patients was performed to strengthen the findings of the main analysis. All tests were two-sided and a *p*-value of less than 0.05 was considered statistically significant. Statistical analyses were performed using R version 4.0 and package “survival” version 3.12 (R Foundation for Statistical Computing, Vienna, Austria) [16].

## 3. Results

### 3.1. Patients

2752 melanoma patients (1310 males and 1442 females; median age: 51 years, IQR: 39–64) were included in the study. Most patients lived in the plain area (2400 patients, 87.3%), followed by 262 patients living near the coast (9.5%), 56 in the hills (2.0%) and 34 in the mountains (1.2%). Demographics and tumor characteristics according to the geographical area of residency are reported in Table 1. Primary site (*p* < 0.0001), Breslow (*p* < 0.0001), ulceration (*p* = 0.03), number of mitoses/mm^2^ (*p* < 00001), and pTNM stage (*p* < 0.0001) were different among patients living in different geographical areas (Table 1). Melanoma of the head/neck region and acral melanoma were more frequent in patients from the hills and the mountains, while melanoma of the trunk and limbs were more common in patients from the plain and the coast. Breslow thickness was higher in patients from the mountains and the hills compared to patients living in the plain area and near the coast. The presence of ulceration and the number of mitoses/mm^2^ were higher in patients coming from the hills. Altogether, patients living in the plain region and the coast presented with an earlier pTNM stage than those from the hills and the mountains.

The sensitivity analysis on 1118 referred patients (766 living in the plain area, 262 near the coast, 56 in the hills and 34 in the mountains) confirmed the differences in terms of primary site (*p* = 0.005), Breslow (*p* < 0.0001), ulceration (*p* = 0.009), number of mitoses/mm^2^ (*p* < 0.0001), and pTNM stage (*p* < 0.0001) among patients living in different geographical areas (Table 2).

### 3.2. Disease-Specific Survival

At a median follow-up of 96 months (IQR 60–132), 523 patients had died (312 from the disease and 211 due to other causes) and 2211 were alive, while information was not available for 18 patients who were lost to follow-up. The five-year disease-specific survival was 83% in patients living in the hills, 92% in those living near the coast, 91% in those living in the mountains, and 91% in those living in the plain area (*p* = 0.10) (Figure 1). In the sensitivity analysis, five-year disease-specific survival was 83% in patients living in the hills, 92% in those living near the coast, 91% in those living in the mountains, and 88% in those living in plain area (*p* = 0.20) (Figure 2). 

Adjusting for imbalanced characteristics at baseline, geographical area and primary site were not associated with disease-specific survival, while higher Breslow (HR 1.07, 95% CI 1.05 to 1.09; *p* < 0.0001), presence of ulceration (HR 2.96, 95% CI 2.25 to 3.89; *p* < 0.0001), higher number of mitoses/mm^2^ (HR 1.07, 95% CI 1.06 to 1.09; *p* < 0.0001), and pTNM III-IV (HR 2.68, 95% CI 2.06 to 3.48; *p* < 0.0001) were identified as risk factors for disease-specific survival (Table 3). The sensitivity analysis on referred patients confirmed such findings (Table 3).

### 3.3. Disease-Free Survival

At the time of the analysis, 393 out of 2732 patients had experienced a disease recurrence, while the information was not available for 20 patients. The five-year disease-free survival was 78% in patients living in the hills, 87% in those living near the coast, 88% in those living in the mountains, and 88% in those living in plain area (*p* = 0.05) (Figure 3).

In the sensitivity analysis, the five-year disease-free survival was 78% in patients living in the hills, 87% in those living near the coast, 88% in those living in the mountains, and 82% in those living in the plain area (*p* = 0.05) (Figure 4). 

Adjusting for imbalanced characteristics at baseline, the geographical area was not associated with disease-free survival, while melanoma in the head/neck vs. trunk (HR 2.03, 95% CI 1.42 to 2.90; *p* < 0.0001), higher Breslow (HR 1.06, 95% CI 1.04 to 1.08; *p* < 0.0001), presence of ulceration (HR 2.28, 95% CI 178 to 2.91; *p* < 0.0001), higher number of mitoses/mm^2^ (HR 1.07, 95% CI 1.05 to 1.08; *p* < 0.0001) and pTNM III-IV (HR 3.50, 95% CI 2.75 to 4.45; *p* < 0.0001) were identified as risk factors for disease-free survival (Table 4). These findings were broadly confirmed by the sensitivity analysis on referred patients (Table 4).

## 4. Discussion

The most important environmental risk factor for developing melanoma is the excessive exposure to UVR. However, the relationship between sun exposure and melanoma is very complex, and is affected by the level and pattern of UVR exposure. In the literature, the measurement of sun exposure is particularly challenging when comparing findings from different investigations, which report heterogeneous methods of recording and coding such information. To our knowledge, no objective approach is currently available for the evaluation of different patterns of exposure, and for the classification of the level of exposure. The geographical area of origin could be an interesting factor which may help to assess the pattern and level of sun exposure in melanoma patients. Previous studies evaluated how the incidence and mortality of melanoma vary according to latitude [17,18,19,20]. Several investigators reported an inverse relationship between latitude of residence and melanoma incidence within populations of similar pigmentation characteristics [21,22,23,24], and an inverse relationship between melanoma mortality and latitude [25,26,27]. Among factors associated with UVR exposure, altitude has been estimated to contribute to 10−12% emission increment for every 1000 m elevation [7]. The Veneto region (Italy) is characterized by a mixed morphology including plain, coastal areas, hills, and mountains; altitude ranges from sea level up to 3383 m above sea level, but there are negligible differences in terms of latitude. Detailed information on morphology and UV exposure are offered by the regional environmental agency [28,29]. Therefore, we believed that the Veneto region might be a suitable area for investigating the relationship between melanoma and altitude. The Veneto Population Registry reports a higher incidence rate of melanoma in the mountain and hill areas when compared to the regional average rate [5]. In this study, we investigated clinicopathological characteristics in a cohort of melanoma patients living in different geographical areas of the Veneto region, and we found significant differences in terms of the primary site of melanoma. Head/neck melanoma was more common in patients from the hills and mountains, which was probably due to the high and intermittent UV exposure of this anatomical district in patients living at higher altitude. On the other hand, melanoma of the trunk and limbs was more common in patients from coastal and plain areas, which may be explained by the more frequent exposure of these anatomical sites in such a subpopulation, thus suggesting different behaviors among people living in different geographical areas. Patients living in the hills and mountains may be less prone to developing melanoma of the trunk and limbs because these body areas are less frequently exposed to the sun, while patients living in plain areas and near the coast have more chances for intense and irregular sun exposure of these sites during the summer. Our findings may suggest a “coast-plain-hill” gradient, with an increasing number of melanomas involving the head/neck site and a decreasing number of melanomas involving the trunk and the limbs. Of note, our data did not show any significant differences in the distribution of malignant melanoma subtypes among geographical areas, while we expected a higher prevalence of lentigo-maligna melanoma in mountains and hills (where the prevalence of head/neck melanoma is higher) and a higher prevalence of superficial-spreading melanoma in plain and coastal areas. To date, few studies have investigated the relationship between melanoma subtype and geographical area of residence [7,8,9,10,11], and we believe that this aspect merits further investigation. In our study, Breslow thickness was higher in patients from the mountains and hills compared to those living in plain areas and near the coast. We believe that the later admission to referral centers might have contributed to such a finding, thus requiring further efforts in improving both screening and informative campaigns in those geographic areas. Furthermore, the presence of ulceration and the number of mitosis mitoses/mm^2^ were higher in patients living in the hills. Overall, patients living in plain areas and near the coast presented with an early pTNM stage than those living in the hills and mountains. These findings may be explained by the higher level of UV exposure and by the intermittent pattern of UV exposure observed in people living in the hills and mountains. It may also be related to the lower vitamin D levels of people living at higher altitude due to reduced chronic sun exposure and a different diet. Low vitamin D seems to be associated to a more aggressive biology of cancer and to a high number of mitoses [30,31,32,33]. When assessing the potential prognostic role of altitude, we found no significant survival differences among patients living in different geographical areas of the Veneto Region, while clinically relevant factors (such as higher Breslow, presence of ulceration, higher number of mitoses per mm^2^ and pTNM III-IV) were confirmed as risk factors for survival. Of note, a recent study found some differences in the expression profile of certain mRNAs and miRNAs with respect to the altitude of residency in patients living in different geographic areas and at different altitudes [34,35]. Since miRNAs are highly regulated by reactive oxygen species, it is possible that different regulatory mechanisms may characterize melanoma at different altitudes due to the different environment and UVR intensity. 

The major limitation of this study is the uneven distribution of patients among geographical areas (coast, plain, hill, mountain). Unfortunately, such a division was retrieved from the classification provided by the Italian Central Statistics Institute, hence we could not apply any meaningful approach to split the “plain area” group (the largest group) into subgroups. In addition, the study used the geographical area of residency rather than the numerical values of the altitudes. Finally, more than 60% of patients lived in the province of Padua, hence we performed a sensitivity analysis in the sub-sample of patients who were referred from other provinces (excluding the province of Padua) to support the findings of the main analysis.

## 5. Conclusions

Our findings suggest a “coast-plain-hill” gradient, with an increasing number of melanomas involving the head/neck site and a decreasing number of melanomas involving the trunk and the lower limbs. The geographical area of residency may contribute to estimate pattern and level of sun exposure in melanoma patients. Further studies including data regarding individual risk factors such as the Fitzpatrick phototype, number of melanocytic nevi, familiar history, and genetic susceptibility are needed to understand the role of altitude in melanoma epidemiology.

## Figures and Tables

**Figure 1 life-12-00745-f001:**
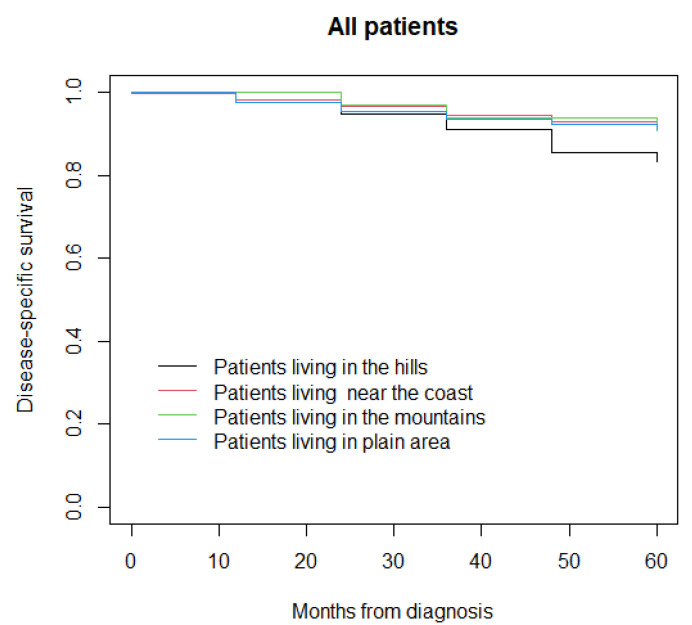
Disease-specific survival in all patients.

**Figure 2 life-12-00745-f002:**
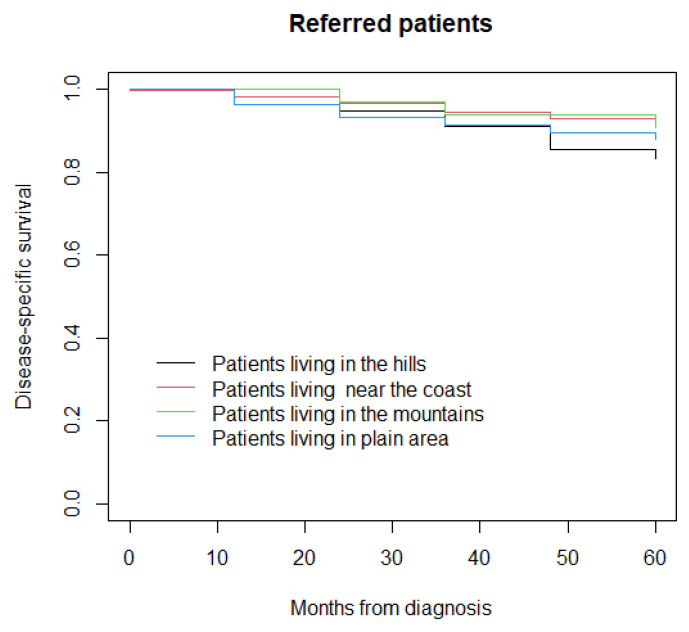
Disease-specific survival in referred patients.

**Figure 3 life-12-00745-f003:**
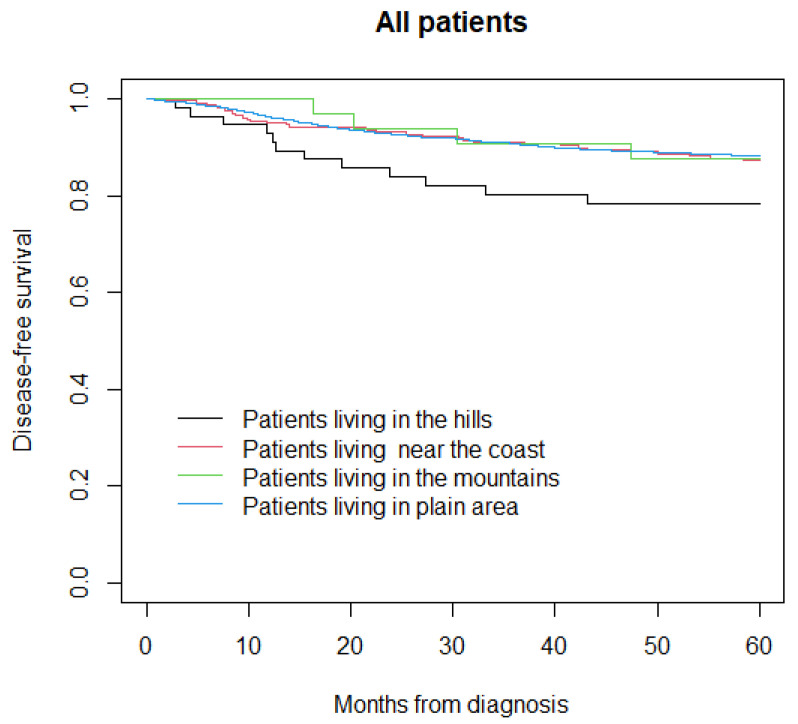
Disease-free survival in all patients.

**Figure 4 life-12-00745-f004:**
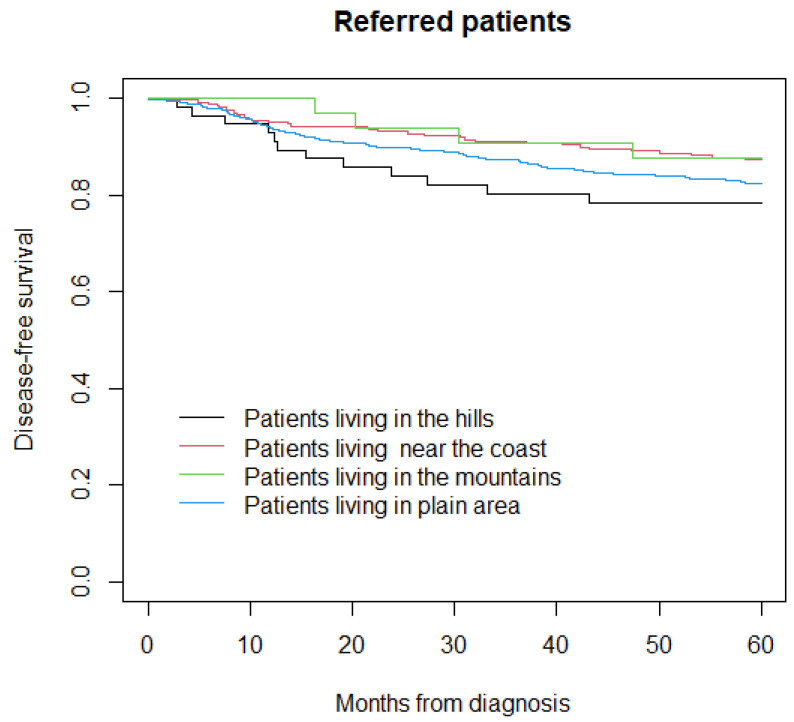
Disease-free survival in referred patients.

**Table 1 life-12-00745-t001:** Demographics and tumor characteristics according to the geographical area of residency of all patients.

	Hills (n = 56)	Coast (n = 262)	Mountains (n = 34)	Plain Area (n = 2400)	*p*-Value
Age, years:					
Median (IQR)	46 (39–63)	53 (39–63)	46 (38–64)	51 (39–64)	0.61
Males: n (%)	29 (51.8%)	127 (48.5%)	21 (61.8%)	1133 (47.2%)	0.34
Primary site: n (%):					<0.0001
Hand and foot	5 (8.9%)	13 (5.0%)	3 (8.8%)	134 (5.6%)	
Head/neck	17 (30.4%)	23 (8.8%)	4 (11.8%)	140 (5.8%)	
Upper limb	6 (10.7%)	35 (13.4%)	4 (11.8%)	346 (14.4%)	
Trunk	17 (30.4%)	120 (45.8%)	14 (41.2%)	1161 (48.4%)	
Lower limb	11 (19.6%)	71 (27.0%)	9 (26.4%)	619 (25.8%)	
Breslow, mm:					<0.0001
Median (IQR)	2.0 (1.1–4.4)	0.7 (0.5–1.8)	1.6 (0.5–2.3)	0.7 (0.4–1.5)	
Data not available	3 (5.3%)	8 (3.0%)	1 (2.9%)	58 (2.4%)	
Ulceration, n (%):					0.03
Absent	33 (59.0%)	206 (78.6%)	27 (79.4%)	1859 (77.4%)	
Present	19 (33.9%)	46 (17.6%)	7 (20.6%)	479 (20.0%)	
Data not available	4 (7.1%)	10 (3.8%)	-	62 (2.6%)	
Mitoses per mm^2^:					<0.0001
Median (IQR)	4 (2–8)	1 (0–3)	2 (0–3)	1 (0–4)	
Data not available	1 (1.7%)	1 (0.4%)	-	13 (0.5%)	
pTNM, n (%):					<0.0001
I	23 (41.1%)	175 (66.8%)	18 (52.9%)	1696 (70.7%)	
II	21 (37.5%)	48 (18.3%)	8 (23.6%)	334 (13.9%)	
III	12 (21.4%)	39 (14.9%)	7 (20.6%)	369 (15.4%)	
IV	0 (0.0%)	0 (0.0%)	1 (2.9%)	1 (0.0%)	
Subtype, n (%):					0.09
ALM	2 (3.6%)	3 (1.1%)	0 (0.0%)	75 (3.1%)	
LMM	2 (3.6%)	5 (1.9%)	0 (0.0%)	32 (1.3%)	
NM	13 (23.2%)	44 (16.8%)	6 (17.6%)	355 (14.8%)	
SSM	32 (57.1%)	192 (73.3%)	26 (76.5%)	1813 (75.5%)	
Other	5 (8.9%)	11 (4.2%)	2 (5.9%)	78 (3.3%)	
Data not available	2 (3.6%)	7 (2.7%)	-	47 (2.0%)	

ALM = acral lentiginous melanoma; LMM = lentigo maligna melanoma; NM = nodular melanoma; SSM = superficial spreading melanoma.

**Table 2 life-12-00745-t002:** Demographics and tumor characteristics according to the geographical area of residency of referred patients.

	Hills (n = 56)	Coast (n = 262)	Mountains (n = 34)	Plain Area (n = 766)	*p*-Value
Age, years:					
Median (IQR)	46 (39–63)	53 (39–63)	46 (38–64)	49 (38–62)	0.43
Males, n (%)	29 (51.8%)	127 (48.5%)	21 (61.8%)	357 (46.6%)	0.32
Primary site, n (%):					0.005
Hand and foot	5 (8.9%)	13 (5.0%)	3 (8.8%)	58 (7.6%)	
Head/neck	17 (30.4%)	23 (8.8%)	4 (11.8%)	76 (9.9%)	
Upper limb	6 (10.7%)	35 (13.4%)	4 (11.8%)	187 (24.4%)	
Trunk	17 (30.4%)	120 (45.8%)	14 (41.2%)	336 (43.9%)	
Lower limb	11 (19.6%)	71 (27.0%)	9 (26.4%)	109 (14.2%)	
Breslow, mm:					<0.0001
Median (IQR)	2.0 (1.1–4.4)	0.7 (0.5–1.8)	1.6 (0.5–2.3)	1.0 (0.5–2.5)	
Data not available	3 (5.3%)	8 (3.0%)	1 (2.9%)	42 (5.4%)	
Ulceration, n (%):					0.009
Absent	33 (59.0%)	206 (78.6%)	27 (79.4%)	539 (70.4%)	
Present	19 (33.9%)	46 (17.6%)	7 (20.6%)	199 (26.0%)	
Data not available	4 (7.1%)	10 (3.8%)	-	28 (3.6%)	
Mitoses per mm^2^:					<0.0001
Median (IQR)	4 (2–8)	1 (0–3)	2 (0–3)	2 (0–4)	
Data not available	1 (1.7%)	1 (0.4%)	-	9 (1.1%)	
pTNM, n (%):					<0.0001
I	23 (41.1%)	175 (66.8%)	18 (52.9%)	435 (56.8%)	
II	21 (37.5%)	48 (18.3%)	8 (23.6%)	153 (20.0%)	
III	12 (21.4%)	39 (14.9%)	7 (20.6%)	177 (23.1%)	
IV	0 (0.0%)	0 (0.0%)	1 (2.9%)	1 (0.1%)	
Subtype, n (%):					0.25
ALM	2 (3.6%)	3 (1.1%)	0 (0.0%)	26 (3.4%)	
LMM	2 (3.6%)	5 (1.9%)	0 (0.0%)	14 (1.7%)	
NM	13 (23.2%)	44 (16.8%)	6 (17.6%)	165 (21.5%)	
SSM	32 (57.1%)	192 (73.3%)	26 (76.5%)	495 (64.6%)	
Other	5 (8.9%)	11 (4.2%)	2 (5.9%)	32 (4.2%)	
Data not available	2 (3.6%)	7 (2.7%)	-	35 (4.6%)	

ALM = acral lentiginous melanoma; LMM = lentigo maligna melanoma; NM = nodular melanoma; SSM = superficial spreading melanoma.

**Table 3 life-12-00745-t003:** Multivariable analysis of predictors of disease-specific survival.

	All Patients		Referred Patients	
	*p*-Value			
	Hazard Ratio (95% Confidence Interval)	*p*-Value	Hazard Ratio (95% Confidence Interval)	*p*-Value
Geographical area:				
Hills	0.95 (0.52 to 1.72)	0.86	1.23 (0.66 to 2.29)	0.51
Coast	1.09 (0.73 to 1.62)	0.68	1.09 (0.73 to 1.62)	0.89
Mountains	0.65 (0.21 to 2.04)	0.46	0.65 (0.21 to 2.04)	0.54
Plain area	Reference		Reference	
Primary site:				
Head/neck	1.46 (0.98 to 2.17)	0.06	1.68 (0.38 to 1.19)	0.18
Trunk	Reference		Reference	
Limbs	0.95 (0.74 to 1.22)	0.71	0.75 (0.52 to 1.07)	0.11
Breslow, mm	1.07 (1.05 to 1.09)	<0.0001	1.13 (1.10 to 1.16)	<0.0001
Ulceration:				0.0006
Absent	Reference		Reference	
Present	2.96 (2.25 to 3.89)	<0.0001	1.95 (1.33 to 2.85)	
Mitoses per mm^2^	1.08 (1.06 to 1.09)	<0.0001	1.06 (1.02 to 1.09)	0.0006
pTNM:				
I-II	Reference		Reference	
III-IV	2.68 (2.06 to 3.48)	<0.0001	2.39 (1.65 to 3.46)	<0.0001

**Table 4 life-12-00745-t004:** Multivariable analysis of predictors of disease-free survival.

	All Patients		Referred Patients	
	*p*-Value			
	Hazard Ratio (95% Confidence Interval)	*p*-Value	Hazard Ratio (95% Confidence Interval)	*p*-Value
Geographical area:				
Hills	0.97 (0.56 to 1.68)	0.91	0.98 (0.55 to 1.73)	0.94
Coast	1.12 (0.78 to 1.60)	0.54	0.95 (0.65 to 1.39)	0.8
Mountains	0.68 (0.25 to 1.84)	0.45	0.62 (0.23 to 1.68)	0.35
Plain area	Reference		Reference	
Primary site:				
Head/neck	2.03 (1.42 to 2.90)	<0.0001	1.56 (0.99 to 2.47)	0.05
Trunk	Reference		Reference	
Limbs	1.22 (0.97 to 1.53)	0.09	1.23 (0.89 to 1.70)	0.21
Breslow, mm	1.06 (1.04 to 1.08)	<0.0001	1.10 (1.07 to 1.13)	<0.0001
Ulceration:				0.004
Absent	Reference		Reference	
Present	2.28 (1.78 to 2.91)	<0.0001	1.63 (1.17 to 2.26)	
Mitoses per mm^2^	1.07 (1.05 to 1.08)	<0.0001	1.06 (1.03 to 1.09)	<0.0001
pTNM:				
I-II	Reference		Reference	
III-IV	3.50 (2.75 to 4.45)	<0.0001	3.20 (2.32 to 4.1)	<0.0001

## Data Availability

The datasets presented in this study can be found in online repositories. The names of the repository/repositories and accession number(s) can be found below: https://zenodo.org/deposit/6511908, https://doi.org/10.5281/zenodo.6511908 (accessed on 2 May 2022).

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
