# Peer review of "Altitude Effect on Cutaneous Melanoma Epidemiology in the Veneto Region (Northern Italy): A Pilot Study"

_life, 2022, doi:10.3390/life12050745_

Round 1

Reviewer 1 Report

In the submitted manuscript Del Fiore et al. explored the association between melanoma and altitude in the Veneto region. They found that head&neck and acral melanomas were more common in patients from the hills and the mountains, while the prevalence of limb and trunk melanomas was higher in patients living in plain and coastal areas. Furthermore, they discovered that with an increase of altitude, the Breslow thickness, ulceration and mitotic rate get worse but no significant difference was observed in overall and disease-free survival. They concluded that geographical area of origin of melanoma patients and the “coast-plain-hill gradient” could help to estimate the influence of different sun exposure and to explain the importance of vitamin D level in skin-cancer control.

This manuscript is quite well written and scientifically sound. However, there are some minor drawbacks, mostly stylistic, which have to be corrected and additionally improved.

1) Line 37: "Earth’s" instead of "Eart’s".

2) Line 45: "disease-free survival" instead of "disease free-survival"

3) Line 58: It is unclear what "host" in the sentence "The epidemiology of melanoma is complex and individual risk depends on host, genetic..." relates to since host's genetics is a risk factor.

4) All numbers larger than 999 should uniformly be written with thousands separator ",".

5) Precisely state which all R-packages (and their version numbers) were used for statistical analyses.

6) In the "Results", provide actual p-values for all mentioned results of statistical analyses.

7) Number (percentage) of "Data not available" samples should be presented in Table 1, not in its footnote. Since majority of data was presented as "n (%)" this should be mentioned in the table, and a remark for those few categories presented as median (IQR) should be eventually put in table's footnote. Also explain what those p-values written in bold present.

8) Regarding Table 2, results of Cox regression, used for assessing the "independence" of biomarkers, are usually presented by juxtaposing tables with the results of univariate and multivariate Cox regression, like for instance in Table III in https://doi.org/10.5301/ijbm.5000291

9) This paper would greatly benefit from a geographical map of the Veneto region, presenting its morphology and, if this data is available, ultraviolet radiation, measured by either average UV index, average number of sunny days or something similar.

Author Response

Point 1: Line 37: "Earth’s" instead of "Eart’s".  

Response 1:  We amended it in the revided version.

Point 2: Line 45: "disease-free survival" instead of "disease free-survival"

Response 2:  We amended it in the revided version.

Point 3 :  Line 58: It is unclear what "host" in the sentence "The epidemiology of melanoma is complex and individual risk depends on host, genetic..." relates to since host's genetics is a risk factor.

Response 3:  We changed “host” with “patient” in the revided version.

Point 4: All numbers larger than 999 should uniformly be written with thousands separator ",".

Response 4:  We amended it in the revided version.

Point 5: Precisely state which all R-packages (and their version numbers) were used for statistical analyses

Response 5: We included such information in the revided version: “Statistical analyses were performed using R version 4.0 and package “survival” version 3.12 (R Foundation for Statistical Computing, Vienna, Austria) [13].” (Methods, page 3).

Point 6:  In the "Results", provide actual p-values for all mentioned results of statistical analyses.

Response 6:  We amended it in the revided version. However, we would like to keep the sentences “The sensitivity analysis on referred patients confirmed such findings (Table 3).” and “These findings were broadly confirmed by the sensitivity analysis on referred patients (Table 4).” to avoid repetitions with the previus lines and to keep a tidy narrative of the results. The mentioned numerical data are offered in the tables immediately after the text.

Point 7:  Number (percentage) of "Data not available" samples should be presented in Table 1, not in its footnote. Since majority of data was presented as "n (%)" this should be mentioned in the table, and a remark for those few categories presented as median (IQR) should be eventually put in table's footnote. Also explain what those p-values written in bold present.

Response 7: We added number and percentage of "Data not available” in the tables. We also added “n (%)” and “median (IQR)” for each variable to improve clarity. In the tables, bold text was a typo during formatting process and was changed in plain text in the revised manuscript.

Point 8:  Regarding Table 2, results of Cox regression, used for assessing the "independence" of biomarkers, are usually presented by juxtaposing tables with the results of univariate and multivariate Cox regression, like for instance in Table III in https://doi.org/10.5301/ijbm.5000291

Response 8: We acknowledge that such presentation approach is often used in the reporting of results in biomedical manuscript. This approach can be useful in studies with limited sample size, because the latter restricts the analytical options for multivariable analysis, hence the authors may be interested in offering some preliminary univariate associations with survival data. Moreover, this approach is sometimes used to justify the choice of the variables to be included in the regression model, although it can be matter of debate in the scientific community (but this falls outside the scope of the current revision). Happily, our study includes a large number of patients which allows the inclusion of all clinically relevant variables in the Cox regression model. Thus, we can directly provide the adjusted estimates of the effects of the variables on survival, which makes the unadjusted estimates from univariate analysis of little interest for the reader.

Point 9: This paper would greatly benefit from a geographical map of the Veneto region, presenting its morphology and, if this data is available, ultraviolet radiation, measured by either average UV index, average number of sunny days or something similar.

Response 9: We agree with the Reviewer about the benefif of a geographical map of the Veneto region. Unfortunately, we could not retrieve any useful map that could be included in the manuscript due to copyright or inadequate information in the map. Hence, we added a link to the environmental agency of the region where the reader can find a informative map of the region and data about UV exposure: “Detailed information on morphology and UV exposure are offered by the regional environmental agency [23,24].” (Discussion, page 10), with two new references (#23 and #24).

Reviewer 2 Report

Dear authors,

An interesting study that can contribute to the literature. I think it will become much more beneficial to the literature with some arrangements.

Some arrangements I suggest;

1. The language of the article needs to be improved (there are grammatical errors and typos). It is recommended that the article is edited by a native English speaker. 

2. It is an important limitation that the numerical values of the altitudes of the cases are not calculated, the limitations of the study should be stated at the end.

3. Could the higher Breslow thickness at high altitudes be related to the later admission to the health centers?

Best wishes...

Author Response

Point 1: The language of the article needs to be improved (there are grammatical errors and typos). It is recommended that the article is edited by a native English speaker. 

Response 1: We revised the language of the manuscript.

Point 2: It is an important limitation that the numerical values of the altitudes of the cases are not calculated, the limitations of the study should be stated at the end.

Response 2: In the study, we classified residential areas in four geographical categories (hill, mountain, plain, coast) based on patient address and according to the Italian Central Statistics Institute (ISTAT). In the revised manuscript, we added the criteria which are used by the the Italian Central Statistics Institute to classify residential areas in Northern Italy: “In Northern Italy, the Institute classifies residential areas above 600 meters as “mountain”, those within 300-600 meters as “hill”, those below 300 meters as “plain” and those with a sea access as “coast”.” (Methods, pages 2-3). We also added a consideration among the limitations of the study: “In addition, the study used the geographical area of residency rather than the numerical values of the altitudes.” (Discussion, page 9).

Point 3 : Could the higher Breslow thickness at high altitudes be related to the later admission to the health centers?

Response 3: We agree with the Reviewer and added a consideration in the revised manuscript: “We believe that later admission to referral centers might have contributed to such finding, thus requiring further efforts in improving both screening and informative campaigns in those geographic areas.” (Discussion, page 10).

Reviewer 3 Report

Article ” Altitude effect on cutaneous melanoma epidemiology in the Veneto Region (Northern Italy): a pilot study”  describes the important issue of the effect of UV radiation on melanoma incidence. Researchers describe cases in a specific location - Veneto Region (Northern Italy).

The topic is very timely and interesting however I have a few comments:

  • Figure 1 and 2 are barely legible and of poor quality please correct this.
  • The four primary study groups are unequal in the number of subjects, e.g. the “mountain group” has 34 subjects and the “plain area” group has 766 subjects. Which makes the analysis difficult and somewhat disturbs the reliability of the study. It would be best to expand the groups.
  • In addition, the bibliography is small and contains few recent articles. This could be enriched.

If the authors respond to the comments in a comprehensive manner, the publication has a chance to be published in this Journal.

Author Response

Point 1: Figure 1 and 2 are barely legible and of poor quality please correct this

Response 1: We agree with the Reviewer, as embedding figures in the original manuscirpt impaired legibility and quality. In the revised manuscript, we included single figures (rather than panel of figures) and enhanced the quality, to improve readability for the reader.

Point 2: The four primary study groups are unequal in the number of subjects, e.g. the “mountain group” has 34 subjects and the “plain area” group has 766 subjects. Which makes the analysis difficult and somewhat disturbs the reliability of the study. It would be best to expand the groups.

Response 2: We agree about the difference in terms of number of patients among the four study groups (which were defined by the area of residency). Unfortunately, such division (coast, plain, hill, mountain) was retrieved from the classification provided by the Italian Central Statistics Institute (ISTAT), hence we have not any meaningful approach to split the “plain area” group (the largest group) in subgroups. We acknowledge such limitation in the Discussion section: “The major limitation of this study is the uneven distribution of patients among geograph-ical areas (coast, plain, hill, mountain). Unfortunately, such division was retrieved from the classification provided by the Italian Central Statistics Institute, hence we could not apply any meaningful approach to split the “plain area” group (the largest group) into subgroups.” (page 9).

Point 3 : In addition, the bibliography is small and contains few recent articles. This could be enriched.

Response 3:  We agree with the Reviewer and added referees listed belowe as requesrt

  1. Keim U, Gandini S, Cutaneous melanoma attributable to UVR exposure in Denmark and Germany. Eur J Cancer. 2021 Dec;159:98-104. doi: 10.1016/j.ejca.2021.09.044. Epub 2021 Nov 3.
  2. Khan AQ, Travers JB, Roles of UVA radiation and DNA damage responses in melanoma pathogenesis. Environ Mol Mutagen. 2018 Jun;59(5):438-460. doi: 10.1002/em.22176. Epub 2018 Feb 21.
  3. Chalada M, Ramlogan-Steel CA, The Impact of Ultraviolet Radiation on the Aetiology and Development of Uveal Melanoma. Cancers (Basel). 2021 Apr 3;13(7):1700. doi: 10.3390/cancers13071700
  4. Buja A, Rugge M, Cutaneous Melanoma in Alpine Population: Incidence Trends and Clinicopathological Profile. Curr Oncol. 2022 Mar 21;29(3):2165-2173. doi: 10.3390/curroncol29030175
  5. Muntyanu A, Savin E, Geographic Variations in Cutaneous Melanoma Distribution in the Russian Federation. Dermatology. 2020;236(6):500-507. doi: 10.1159/000507617
  6. http://geomap.arpa.veneto.it/maps/51#license-more-above (accessed on 2 May 2022)
  7. https://www.arpa.veneto.it/temi-ambientali/agenti-fisici/radiazioni-uv/dati-in-diretta (accessed on 2 May 2022)
  8. De Smedt J, Van Kelst S, Determinants of 25-hydroxyvitamin D Status in a Cutaneous Melanoma Population. Acta Derm Venereol. 2022 Apr 8;102:adv00692. doi: 10.2340/actadv.v102.262.
  9. Lombardo M, Vigezzi A, Role of vitamin D serum levels in prevention of primary and recurrent melanoma. Sci Rep. 2021 Mar 12;11(1):5815. doi: 10.1038/s41598-021-85294-3
  10. Gencia I, Baderca F, A preliminary study of microRNA expression in different types of primary melanoma. Bosn J Basic Med Sci. 2020 May 1;20(2):197-208. doi: 10.17305/bjbms.2019.4271

Round 2

Reviewer 2 Report

Dear authors,

I see that suggested changes have been carried out in totality.

Best wishes...

Reviewer 3 Report

The authors have answered all the questions, therefore I think that the article can be published.